# Release of the Fourth Season of Money Heist: Analysis of Its Social Audience on Twitter durin Lockdown in Spain

**Carmen Cristófol Rodríguez** [1] , **Paula Meliveo Nogués** [2] **and Francisco Javier Cristòfol** [3,*]

1    Faculty of Business and Communication, International University of La Rioja, 26006 Logroño, Spain; carmen.cristofol@unir.net
2    Department of Audiovisual Communication and Advertising, University Malaga, 29013 Malaga, Spain; paulameliveo@uma.es
3    Department of Market Research and Quantitative Methods, ESIC, Business and Marketing School, 28223 Pozuelo de Alarcón (Madrid), Spain
*    Correspondence: fjcristofol@esic.edu; Tel.: +34-606-83-77-78

**Abstract:** Nowadays we are witnessing a significant change in content consumption. This, together with the global health situation, has caused some behaviors to accelerate. This research focuses on the specific case of the lockdown in Spain and the coincidence with the launch of the fourth season of Money Heist compared to the launch of season three. Starting with a review of the theoretical framework, in which the related concepts of coronavirus, television, and Video on Demand (VOD) platforms are presented, the importance of transmedia communication is also introduced. The methodological aspect is developed through content analysis and in-depth interviews. The tool used on the first methodology has been Twlets. With regard to the sources, the specific bibliography of the audiovisual sector, the official profile of the series on Twitter and personal interviews with professionals from the communication department of the production company, Vancouver Media, and from the series directing were taken into account. The methodology used to carry out this work has been the analysis of quantitative–qualitative content of the various sources consulted. The results of the study are presented in graphs, crossing the data from the different sources to detect the strategies of marketing and communication used for the release of the fourth season of the series. These results reflect the change in the communication strategy, the behavior of the social audience of the Twitter account of Money Heist (La Casa de Papel) and its relationship with the period of lockdown in Spain.

**Keywords:** Netflix; Covid-19; twitter; Money Heist; audiences; series; VOD; release

## 1. Introduction

The lockdown in Spain implied an acceleration in the consumption of Video on Demand (VOD) content. Throughout the theoretical development it will be shown how the effect of the coronavirus in its concrete relation with the Money Heist series has meant a relevant increase with respect to the season premiered before knowing the influence of the virus in the world.

The main objective of this research is to value the impact of lockdown due to Covid-19 on the release of the fourth season of the fiction series Money Heist.

The specific objectives of this analysis are:

- Comparing the release in social media of seasons 3 and 4 of the series in question.
- Quantifying the number of tweets, likes, and replies of Money Heist account during the first month after the release of the third and fourth seasons of the series.

- Analyzing the audiovisual content of these tweets.
- Identifying the most used keywords.
- Identifying the most commonly used keywords that directly or indirectly refer to the lockdown.

This paper has a structure that is organized in four points. Firstly, a theoretical framework is presented in which previously published studies and papers are collected, at the end of this section, objectives are shown. On the next section, we present the methodology. The results are presented below, and this research is concluded with the discussion and final conclusions.

## 2. Theoretical Background

### 2.1. Coronavirus, Television, and Platforms

The health crisis caused by the Covid-19 pandemic, as well as the resulting lockdown, have changed the television consumption habits of viewers in Spain during the state of alarm, which lasted from March 14 to 20 June 2020. During these fourteen weeks, each individual consumed an average of 278 min of television per day, which is almost an hour more (51 min) of television per day compared to the same period last year [1].

As can be deduced from this data, during the period in which lockdown was an imposed reality, television consumption increased. According to Martínez [2]:"Thanks to the media, today's societies are in permanent communication. Therefore, they are a persuasive tool that allows us to keep in continuous communication during any event" (p. 74).

Martínez [2] continues by stating that "media, on the one hand, contribute to the formation of more educated, better informed and more free individuals; but, on the other hand, they can serve for the diffusion of a superficial, routine, consumerist culture, which alienates us with false lures" (pp. 74–75).

In 2019, each viewer in Spain consumed an average of 227 min per day. April 2020 was the month of maximum television consumption in the history of records in this country. The average was 302 min of daily consumption per person. According to Barlovento [1], "March, with only fifteen days of lockdown, occupies the second place in this historical ranking with 284 min of television per person and day".

During the week of 16–22 March 2020, the first week of lockdown in Spain, a daily television consumption per person of 325 min (five and a half hours) was recorded, which fell considerably fourteen weeks later, when de-escalation began. In other words, consumption decreased when the population was allowed to go out [1].

Regarding Video on Demand (VOD), HBO registered the highest growth, followed by Spanish VOD platforms like Filmin, Movistar Plus, or international like Netflix and Amazon Prime Video. Streaming content consumption rose by an average of 201% since the start of the quarantine in Spain. Although Netflix did not register such growth during the lockdown, it was the most viewed platform. Blanes (2020) states that "this is because it is the platform with the greatest support in Spain before the crisis, making it more difficult to grow at a fast pace". During the period of lockdown in Spain, Netflix released Elite and Money Heist, the latter being an international phenomenon and "one of the most viewed products in the world" [3].

The premiere of the fourth season of Money Heist on Netflix was scheduled for April 3. It was scheduled before the health crisis caused by the Covid-19 pandemic began in Spain. The communications department of the series' producer, Vancouver Media, confirms that the release of this new season was planned for that date without any knowledge of the possibility that the population of Spain might be confined by government order. A coincidence that has favored visualizations [4].

In contrast, we find the case of Movistar Plus. Its strategy consisted of modifying its programming in order to bring forward the premiere of a series and offer services adapted to the moment. Despite the success of fiction series broadcast on VOD platforms, due to the high risk of virus infection, filming in Spain and other affected countries had to be suspended for a certain period of time by government

order to protect both the artistic and technical teams working on them. "This is another reason for the news to be so prevalent, since it only takes one anchor and one cameraperson to present the news (in fact, many anchors do it from the comfort of their homes)" [5] (pp. 74–75).

### 2.2. Netflix and Fiction Series in Spain

The consumption habits of audiovisual products have changed for several generations because of the integration of the Internet into the daily life of users, the technological renewal, and the emergence of VOD platforms. For Ojer and Canapé [6], "the development of the Internet, the implementation of mobile devices and the new audiovisual consumption habits have led to the creation of new business models in the exploitation of audiovisual products".

Taking Netflix or Filmin as examples, customers can watch their favorite movies and series whenever and wherever they want. These companies also face challenges such as piracy and Internet connection speed, as well as the conditions that each country and production companies set on the rights to broadcast audiovisual content. In the case of Netflix, the number of subscribers has not stopped [6]. In Spain, 53.8% of households are subscribed to some form of pay television. The sum of traditional and streaming pay television exceeds for the first time ten million households throughout the country [7].

According to Barlovento [7], Netflix exceeds 14.1 million people in Spain who receive its contents (whether they are subscribers or not), which would correspond to 6.2 million households. Of this total figure, the majority are women, specifically 52%. The largest age group is adults between 35 and 44 years, representing 21% of the total number of viewers. As shown in Figure 1.

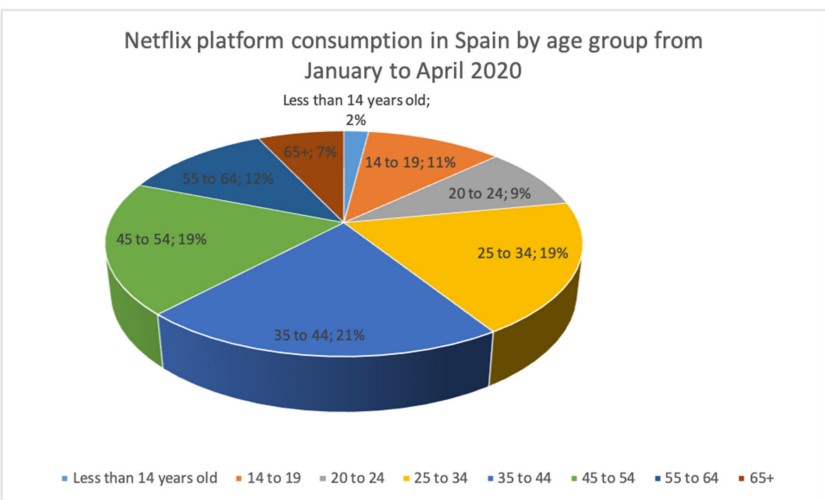

**Figure 1.** Netflix platform consumption in Spain according to age group, from January to April 2020. Source: Data extracted from EGM (Estudio General de Medios) [8].

The affordable and comfortable accessibility of users to platforms and specifically to Netflix, favors the increase in consumption of its extensive catalogue, including fiction series, thus benefiting the growth of production. On 31 March 2016, Netflix announced in an official statement the creation and development of its first original series filmed entirely in Spain, which would be released exclusively on this platform worldwide.

The project is commissioned to Bambú Producciones, directed by Ramón Campos and Teresa Fernández Valdés, producers of the hit series Grand Hotel and Velvet. The platform initially signed up for sixteen episodes of the series Cable Girls, agreeing on the length of each one in 50 min. After broadcasting the first season, Netflix commissioned the production company to produce more episodes of this fiction. Cable Girls premiered on 28 April 2017 in Spain with considerable success in

this country and in other parts of the world. At this time, there are three seasons available on Netflix (with a fourth on the way) [9].

After the international success of this project, Netflix begins to release other fictional series created and produced in Spain, such as Money Heist (Vancouver Media, 2017), Elite (Zeta Ficción TV, 2018), High Seas (Bambú Producciones, 2019), the second and third seasons of Paquita Salas (Apache Films and Suma Latina, 2019), Hache (Weekend Studio, 2019), and Three Days of Christmas (Filmax and Netflix, 2019). This increase in the production of series in Spain for broadcasting on platforms is associated with a qualitative change in the way they are designed, that is, in terms of form and content. This is how Javier Quintas, one of the directors of Money Heist, explains it: "Broadcasting the series on a platform like Netflix has allowed us to introduce plots, characters and dialogues without having to restrict the series to the tastes of a heterogeneous audience" [10].

In the same vein, Quintas [10] states that: "Previously, fictions had to be designed and broadcast on channels that needed to reach all members of the family. Thanks to the platforms, that's over. The way of doing things has also changed, of course. It is an evolution towards creativity".

For industry executives, VOD platforms represent a new possibility to explore and develop new ways of creating, developing, and making stories for the audiovisual market. In Carrillo Bernal's work [11], producer Javier Olivares defines the phenomenon as a new paradigm, explaining how the relationship between the spectator and the platform is produced.

## 2.3. Money Heist

Money Heist is a fictional television series with four seasons recorded and broadcast. It is a Vancouver Media production initially for Antena 3 in 2017 [12], where the first and second seasons were shown. Afterwards, this fiction series continued on Netflix.

The series is produced by Vancouver Media, the company created and directed by Álex Pina, who after leaving Globomedia (for which he has performed the functions of executive producer, scriptwriter and creator of fictions, such as Periodistas (Journalists), Los Serrano (The Serrano), Los hombres de Paco (Paco's Men), El barco (The Ship), andLocked Up) created this new factory supported by a team with solid experience in the audiovisual sector. A list of credits is shown in Table 1.

**Table 1.** List of credits Money Heist.

| List of Credits | |
|---|---|
| Executive Production | Jesús Colmenar and Cristina López Ferraz |
| Co-production | Javier Gómez Santander and Migue Amoedo |
| Directors | Jesús Colmenar, Koldo Serra, Álex Rodrigo and Javier Quintas |
| Writers | Luis Moya, Juan Salvador López, Ana Boyero, Emilio Díez, Esther Morales, Esther Martínez Lobato, Javier Gómez Santander and Álex Pina, coordinated by Javier Gómez Santander |
| Visual concept and directors of photography | Migue Amoedo, David Azcano and Sergi Bartroli |
| Cast | Úrsula Corberó, Álvaro Morte, Itziar Ituño, Pedro Alonso, Alba Flores, Miguel Herrán, Jaime Lorente, Esther Acebo, Enrique Arce, Darko Peric, Hovik Keuchkerian, Luka Perós, Belén Cuesta, Fernando Cayo and Rodrigo de la Serna, among others |

Source: Vancouver Media, 2020.

According to Rebollo-Bueno [13], "Money Heist is presented as a cultural product that criticizes the capitalist system, the ruling class, thanks to a change in roles". In this case, the politicians are pointed out as the "bad ones" and the thieves as the "good ones".

The international phenomenon that the series created by Alex Pina has entailed, has been possible thanks to the powerful reception that is not only reflected in the visualizations of its episodes on Netflix, but also in the remarkable number of impressions that Internet users wrote on YouTube about the release of the trailer for the fourth season. The impact on the audience of this teaser on the Internet has been quantitatively greater than that caused by Game of Thrones trailer.

According to Parrot Analytics, quoted by *Observer*, the fourth season of Money Heist is 31.73 times more in demand than the average series around the world and beat popular series such as Game of Thrones, The Walking Dead, Brooklyn Nine-Nine, and Westworld. Demand for the fourth season during the first three days has jumped 36.6 percent compared to the third season [14].

About the opacity of Netflix data [15] states that "it will be heard everywhere that 65 million households around the world have started to watch the fourth season of Money Heist". He also questions that data because, according to the author, "Netflix really does not indicate the number of people who watch the series", but the number of households that play the first two min of it.

The Ibero-American Observatory of Television Fiction 2019 ([16], within its ranking of national and Ibero-American fictions exhibited in VOD in 2018, places Money Heist in first position in Chile and Mexico, third in Argentina, seventh in Brazil, ninth in the USA, and tenth in Uruguay.

OBITEL also states, about the success of the series in Brazil, that "it has been demonstrated by the number of tweets and impressions in this social media: more than 120 million impressions and more than 1 million posts" and, about the fiction series in Spain, states that "the winner of the first Emmy awarded to a Spanish fiction has become the first Spanish series to inspire a line of dolls (Funko Pop!)" [16].

The international success of the series is not only reflected in its television audiences and its impact on social media, but also from an international academic point of view, it is worth noting the amount of research published in the form of articles, conferences or theses, even in various languages, which address the series (Spanish, English, Portuguese, and Italian). The countries of origin of these publications are Spain, retrieved from MDPI, Scopus and Google Scholar, [12] Colombia [17], Argentina [18,19], Ecuador [20], USA [21], Italy [22], Brazil [23–25], Greece, and UK [26].

Furthermore, in order to know the level of fidelity of the target at which any series is aimed, new tools have been created such as Naive Bayes, a system of quantitative analysis that has confirmed that the majority of the impressions that viewers made about the trailer of the fourth season of Money Heist are positive, and that the demand of the audience is significant [13].

Money Heist has become one of the most binge-watched series, which means that viewers consume this product with a certain need to satisfy their curiosity to know what will happen in the episodes they have not yet watched. According to Castelló [27], "this individualistic and binge-watching consumption is one of the defining features of video on demand content".

The identification with the main characters designed in a complex and morally ambiguous way, the chemistry between them, the drawing of the empowerment of the female characters, and the challenge they pose to the corrupt power have provoked such empathy in the followers of the series that it has even been reflected in social movements of the contemporary age, as it happens with Bella Ciao [27].

The significance of this production has gone beyond the mere entertainment of the viewers, legitimizing the latent desire of the audience to challenge the corrupt power that today controls and abuses the population in Spain and in other countries. Episode after episode, the series offers a critical view of the system disapproved by the authorities and the power elites who try to indoctrinate the population to assume their condition as an exploited class and obey.

The third season premiered on Netflix on 19 July 2019. The fourth, on 4 April 2020, during the lockdown in Spain because of the global epidemic of Covid-19. The Vancouver Media Communications Department confirms that "the release date of the fourth season was set many months in advance of the epidemic and the government decision to activate the state of alarm did not change it" [4].

*2.4. The Use of the Internet and Social Media during Lockdown*

From 13 March 2020 until 21 June Spain has been immersed in a state of alarm decreed by the Government of Spain after its approval in the Congress of Deputies, due to the pandemic produced by the expansion of the Covid-19.

During this period and in different stages, the population has been forced to confine itself at home, which has circumscribed its scope of direct relationship to its circle of cohabitation. It is in this context that the use of digital social media has been strengthened as the main means of socialization. WhatsApp, Facebook, Twitter, Instagram, or TikTok have become the tools for contact with the old reality: video calls, live broadcasts, or the use of Twitter instead of the radio as a means of more immediate information have multiplied. With the disappearance of leisure places, digital social media have become them: yoga classes, cooking classes, or investment classes for the stock market, have been intermingled with teleworking, home cinema or virtual family trivia. An example of this is the data provided by Telefónica, in which it reports that, from 13 March to 13 April, it recorded Internet traffic equivalent to that of all of 2019 [28].

Until March 13, the average use of social media was three h and fifteen min a day on the Internet. During the lockdown, according to the Spanish operators (Telefónica, Orange, Vodafone, MásMóvil, and Euskaltel), the general traffic has experienced increases close to 40%, while the use of mobiles has increased around 50% in voice and 25% in data.

According to the report "Digital Consumer 24 h Indoors" [29], during the period of lockdown, Spanish people have spent two more hours a week consuming social media, 13 h a week and 79 h surfing the Internet during the first six weeks. Before March 13, the average was 11.3 h per week.

Interestingly, this 15% increase in use was mainly devoted to consuming third-party content rather than producing their own, reflecting the lack of creativity or slackness of the early days, when there was still no habit of being at home.

According to this same study, the consumption of films and series—via platforms—has grown from 53% to 72%, making evident the increase in leisure time that the population spends at home and, therefore, a greater demand for audiovisual content. Netflix keeps having the largest number of users with a 75% of the total, although during lockdown, Amazon Prime Video followed by Movistar + and HBO, have registered considerable increases, in that order.

According to [29], the consumption of total hours of films and series online has increased with respect to 2019 from 38.5 h per week to 45.6 h, 18% higher. Maira Barcellos in Nielsen [29] states that "the lockdown of Spanish population at home has led to greater use of the Internet" and it has changed the concept of buying channel for the consumption of information and entertainment.

According to the study of the application for the management of parental control Qustodio after lockdown, children and adolescents up to 16 years have increased by 170% the use of social media and spend 80% more time connected to the Internet [30].

Following Cachia et al. [31], "the sheer volume of user-generated content available on social networks allows for sophisticated environmental scanning through data mining". Mount and Garcia Martínez [32] concluded that "collective intelligence also helps reduce cognitive bias by allowing users to focus on processes, problems, and solutions that occur naturally".

Twitter During the Lockdown

Social network audiences have been unequivocally increased by the arrival of the Covid-19, not only because of the constant need to be kept informed but also because of the increased time that can be dedicated to them. Several studies have been conducted to analyze the audiences of Twitter during the pandemic, most notably that of Trajkova et al. [33], which analyzes the perception of data visualization through this network; that of Li et al. [34], which conducts a content analysis of tweets published to study the stigmatization of the virus; Gencoglu & Gruber [35] attempt to infer and identify the variables that affect public attention and sentiment through twitter messages.

In a general idea of social media, the research carried out by de las Heras et al. [36] monitors the digital ecosystem during March and April 2020, specifying how it has affected risk communication in uncertain contexts and its impact on the emotions and feelings derived from semantic analysis in Spanish society during the COVID-19 pandemic.

*2.5. Transmedia Communication Strategy*

Social networks help organizations to approach their audiences more directly and closely than through any other communication channel. It is a fact that they also force themselves to be present on Twitter, the social network of dialogue par excellence, but it is also true that this is the most demanding social network, since to participate in it is essential to have a very proactive presence [37].

However, despite many benefits that this network can bring, it is still common to see that how the communication strategies developed by organizations are still not completely effective, as shown in the academic literature, where recommendations and advice abound [38–43].

Transmediation in the case of LCDP is one of the strategies mostly used by this series, which through the analysis developed evidences that the content of the messages goes far beyond the merely television, creating transmediatic universes through mobile apps as described by Navarro et al. [44]. Or all the merchandising described by Formoso et al. [45]. But without a doubt, twitter becomes the key tool in the transmediatic universe of the series, as can be seen in the tweet analysis developed: the characters leave their own fiction to acquire an independent personality, as is the case of The Professor and his verified twitter account (@elprofesor).

## 3. Materials and Methods

The methodology used was the content analysis of the tweets posted by the Twitter profile @LaCasaDePapelTV, the official account of the series. This profile was created in November 2016 and as of July 2020 has about 431,000 followers.

To evaluate the impact of pandemic lockdown at the release of season 4, tweets posted between 3 April and 3 May 2020 have been analysed and compared with those posted during the first month after the release of season 3 July 2019 and 19 August 2019.

The application Twlets has been used to export the content of the tweets to Excel. This application was chosen because it allows to extract the content of the analysed profile, including the following items, which are fundamental to achieve the objectives of this research:

- ID: it indicates the URL of the tweet.
- screen_name: it indicates the username of the account that posted the tweet.
- created_at: it indicates the date and time of posting of the tweet.
- fav: it indicates the number of times the tweet was marked as a favorite.
- rt: it indicates the number of times the tweet was retweeted.
- RTed: it indicates if the post is a retweet from another account and, if so, indicates from which account.
- text: it shows the text of the post, including hashtags and URLs.
- media: it shows up to four columns, with the URLs of pictures and videos attached to the post.
- Sentiment: it is divided into four columns where an algorithm classifies the messages, depending on if they are positive, negative, or neutral, and adds them a value named Compound.

The analysis of sentiments is carried out by means of an algorithm called Valence Aware Dictionary and Sentiment Reasoner (VADER), defined by Hutto and Gilbert [46]. According to Rayarel [47], VADER is a reliable method to calculate the polarity of tweets since it is frequently used to analyze social media. VADER has a dictionary of social media vocabulary and matches the words in the tweets to this dictionary. It assigns a composite value to the tweets and categorizes them as positive, negative, and neutral. It is shown on the Figure 2 as a flowchart.

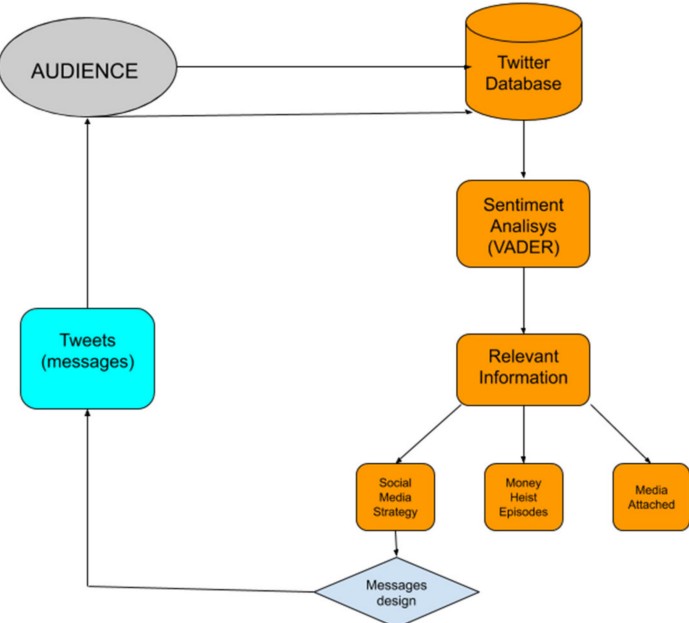

**Figure 2.** Own elaboration. Addapted from: Zamarreño et al. [39].

The analysis sheet that has been used contains the following variables:

- Date of tweet
- Number of likes
- Number of retweets
- Number of replies
- Full text
- Number of multimedia links
- Keywords referring to lockdown.
- Content of tweet: If it is a retweet, if it is about the content of the series, about the characters, if it is advertising.
- Emotional component: positive, negative, and neutral

On the other hand, the methodology of the in-depth interview has been chosen for the compilation of qualitative data. For Taylor and Bogan [48], the interview is a set of repeated encounters between an interviewer and one or more interviewees that is aimed at understanding the interviewee's vision of their experience. In this case, the interview is based on a script.

For Olaz [49], this script has a clear and simple objective: to get as much information as possible about the research question. Rodríguez Tarodo [50] defines this guide as a structured protocol, but not a closed one, since it deals with generic areas. In this case, an in-depth interview has been conducted with Sara Solomando, documentary filmmaker and communications director of Vancouver Media for Money Heist. Solomando has a degree in journalism, specializing in radio and television, and according to the newspaper Hoy [51] "Solomando is the first person who receives the script and has also made a cameo in several episodes".

Javier Quintas (one of the directors of the series Money Heist) was also interviewed via videoconference and he answered eighteen open questions. The details regarding his answers were clarified during that same conversation, which took place on 30 April 2020 and lasted one hour and fifteen minutes.

## 4. Results

The data extraction has been carried out with the Twlets tool. From the dump of these raw data, the analysis has been done with the tool Statistical Package for the Social Sciences (SPSS). Mainly, the results have been processed by the authors in terms of the theories and audience studies mentioned in the theoretical framework.

### 4.1. Results of Content Analysis

The data was obtained on 7 July 2020 through the application Twlets. Below, on Table 2, is a summary of the most relevant results:

**Table 2.** Summary of the results of content analysis of Money Heist's Twitter profile.

|  | S3 | S4 |
| --- | --- | --- |
| **Total of tweets** | 62 | 100 |
| Average tweets per day | 2 | 3 |
| **Likes** | 18.735 | 111.243 |
| Average likes per tweet | 302 | 1.112 |
| Average likes per day | 625 | 3.708 |
| **RT** | 1.846 | 14.292 |
| Average RT per tweets | 30 | 143 |
| Average RT per day | 62 | 476 |
| **Replies** | 652 | 569 |
| Average replies per tweet | 10 | 6 |
| Average replies per day | 21 | 19 |

Source: Own elaboration.

Season 4 of Money Heist meant for the Twitter profile to increase its presence in this social network by 38 posts, with an average of 50% more posts compared to season 3. Although these changes were numerically not very relevant in view of their comparison, the big increase was in the interactions in general.

Twitter users marked 111,243 times as favorites the 100 posts of @lacasadepapel, which compared to the third season, mean 109,508 times more. If we focus on the retweets, from the 1846 made to the 62 tweets posts in the release period of the third season, it increased to 14,292 in the release period of the fourth season, an increase of 12,446. It is in the replies in the only section in which the third season has better results than the fourth: 652 against 569. A difference of 83 replies.

In season 3 we observe that 62 tweets are posted in the selected period, of which only 38 include the hashtag #LaCasaDePapel, the rest are promotion for Atresmedia and other of its series. These 38 tweets add up to a total of 16,075 favorites and 1488 retweets. Of those that include the hashtag of the series, only 3 attach a link to multimedia content. These three tweets add up to 1930 favorites and 349 retweets.

As seen in Table 3, the use of the hashtag #LaCasaDePapel in the messages sent by the official verified Twitter account has doubled: from 38 in the period studied for the third season to 76 in the period of the fourth. The number of retweets made by the account has also increased significantly, from 5 to 27, 22 more posts between the two periods of analysis. Money Heist's Twitter profile makes a total of five retweets during this period: two from ATRESplayer, two from antena3.com and one from Flooxer. These retweets do not get marked as a favorite in any occasion, but they do get a total of 247 retweets and 4 replies.

**Table 3.** Summary of the content analysis of the Twitter profile of *Money Heist.*

|  | S3 | S4 |
|---|---|---|
| #LaCasaDePapel | 38 | 76 |
| Retweets | 5 | 27 |
| Multimedia | 3 | 26 |
| Words referring to pandemic | 0 | 13 |

Source: Own elaboration.

As far as the hypertext attached to the tweets is concerned, we find that it is multiplied by 8.6; while in the period of the third season five multimedia elements are posted, in the case of the fourth season up to 26 multimedia elements are attached to the tweets posted by @lacasadepapel. Finally, in the space for the promotion of the third season there is no mention of the pandemic, since there was no such thing, but in the fourth season terms related to the pandemic are used, such as #yomecorono, coronavirus or lockdown.

Asseen on Table 4, for the VADER analysis, we can find three levels, being −1 extremely negative, 0 neutral and 1 extremely positive the tone of the tweets. In this sense, the posts related to season 3 have a slightly negative tone, although as it is in both seasons close to 0, it is considered that in any case they are neutral posts.

**Table 4.** Valence Aware Dictionary and Sentiment Reasoner (VADER) analysis.

|  | Compound | Negative | Neutral | Positive |
|---|---|---|---|---|
| S3 | −0.073 | 0.043 | 0.938 | 0.019 |
| S4 | 0.0005 | 0.019 | 0.963 | 0.018 |

Source: Own elaboration.

On Table 5 is shown that the length of the tweets has also been analysed, which varies from twelve words in the third season to twenty-one in the fourth.

**Table 5.** Most used words in the tweets about season 3 and 4.

| SEASON 3 | | SEASON 4 | |
|---|---|---|---|
| **Used Words** | **Repetitions** | **Used Words** | **Repetitions** |
| Lacasadepapel | 38 | Lacasadepapel | 65 |
| Tokio | 11 | Series | 14 |
| Enjoy | 9 | Professor | 11 |
| Character | 7 | Robbery | 10 |
| Robbery | 7 | Tokio | 8 |
| Professor | 7 | Episode | 8 |
| Series | 6 | Denver | 8 |
| Season | 6 | Plan | 8 |
| Río | 5 | Coronavirus | 7 |
| Úrsula | 4 | Río | 6 |
| Finale | 4 | Úrsula | 5 |

Source: Own elaboration.

*4.2. Results of the Interviews*

The personal interview conducted with Sara Solomando (Vancouver Media's Communications Department) on 20 June 2020, consisted of seven initial open-ended questions that were increased by four more to clarify details. The interview was conducted in writing via email. The questions focused on determining the exact date of the release of the third and fourth seasons of Money Heist, determining whether the decision to release the new season during the lockdown was premeditated or accidental, and obtaining information about the promotion strategy that the production company had followed, especially on the social network Twitter.

Solomando accurately dated the third and fourth season premieres on 19 July 2019 and 4 April 2020, respectively. He confirmed that the fact that the release of the new season coincided with the lockdown in Spain due to the Covid-19 pandemic was coincidental as it was planned months before the contagion alert arrived in Spain, and that social media "play an essential role in promoting the series". He added that it is important not to fear spoilers or negative criticism because that would lead to total inactivity. With respect to the communication strategy, he commented the following:

The communication strategies are elaborated by the Netflix communication and marketing teams and go through an intense activity in social media, as well as behind the scenes material recorded with the actors and actresses and the design of the promotional campaign with the traditional media (blogs, written press, radio and television) (S. Solomando, personal communication, 20 June 2020).

The essential difference in the promotion of the fourth season of Money Heist with respect to the third was "the impossibility of conducting face-to-face meetings and interviews, so the press junket has been replaced by video conferences and calls".

The interview with Javier Quintas (one of the directors of the series Money Heist) was conducted through a videoconference during which he answered eighteen open questions. The details regarding his answers were clarified during that same conversation, which took place on 30 April 2020, and lasted one h and fifteen min. The questions focused on obtaining information about the creative and productive novelties offered by broadcasting a fiction series on an international digital platform such as Netflix, among others. His answers were oriented to the positive evaluation of the possibilities of the new platforms, in the sense of being able to include the fragmented audience in each of the countries in which it operates. By adding these specific audiences in each country, the target of the series multiplies, which gave writers and directors great creative freedom, since the product did not need to please all members of the family to be profitable.

## 5. Discussion

With respect to the objectives set out in this research, according to the results obtained, there are remarkable differences in the communication strategies in the social media profiles of the series Money Heist with respect to the release of seasons 3 and 4. In that sense, Sara Solomando (Communication Department of the producer of this fiction, Vancouver Media) has pointed out the existing influence of the lockdown due to the Covid-19 pandemic in an evident change of behavior of the social audience with respect to the period corresponding to the launch of the third season. Noteworthy differences can be observed in the content posted: in the third season, the main objective was to increase the number of followers, with calls to action to get replies, retweets, and likes with shorter and more direct tweets, while in the fourth season, the strategy has been aimed at attaching elements, as followers have had more time to devote to social media during lockdown. It is also detected the growth of retweets in this fourth season to take advantage of the number of followers of other fiction series and promote other new works such as Mentiras and Veneno, broadcast in Atresmedia.

The second objective was to quantify the number of tweets, likes, and replies to @lacasadepapel profile during the first month after the release of the third and fourth seasons of the series. Thanks to such quantification, it is easy to observe the increase in interactivity with users, which reinforced by the reports of Nielsen [29] or Moreno [52] reflect the increase in time that Spanish people have spent surfing the net, either consuming fiction on demand, or consulting and interacting in social media.

This social audience is more interested in consuming new content from each plot of the series than in doing so in high quality, which increases the attraction to the official profiles of the series (specifically Twitter for the technical possibilities that it offers).

The third objective, analyzing the audiovisual content of the mentioned tweets, also reflects how the social media manager of the series has known how to take advantage of the possibilities derived from the situation of lockdown to introduce a greater number of audiovisual or multimedia elements in this new release, making the most of the technical and planning options available in order to achieve greater engagement with the audience. This level of engagement is also reflected in the neutral tone of the posts produced, which is shown in VADER analysis.

It is relevant to see how, despite all the above, the most used keywords in both seasons are the same and only the word "coronavirus" is added to the top of them. Nevertheless, although with less repetitions, other words appear referring to the situation, such as "pandemic", "quarantine", "lockdown", and "video call". With respect to the audiovisual content of the production shown in the network to comply with the marked promotion strategy, a greater development and spreading of creativity can be perceived, also referred to by Javier Quintas [10] in the scriptwriting and production of the work. Social media exhibit pieces that sometimes reveal plot content, and this works to capture the viewers who want to satisfy their curiosity about how the story ends. This, together with the core idea that Sara Solomando [4] told us about not being afraid of making spoilers, leads to a greater interest in the audience to consume more pictures of the series shown in the media. It also leads to an interest to see the work, positively influencing the actions of feedback on Twitter, since the consumer acts on impulse at times, guided by the sensations and emotions awakened by the narrative and the characters, as the networks are revealing new details about the latest season.

## 6. Conclusions

We can therefore deduce that transmedia storytelling has become a powerful communication strategy in the release of the fourth season of the series. It is true that nothing was known about the lockdown when the series was scheduled to be release, but the strategy used in social media by the community manager is remarkable. According to Jenkins, Ford and Green [53] transmedia storytelling is the new aesthetic around media convergence. It is the possibility of generating worlds and their convergence. Therefore, it is relevant to conclude that the transmedia work from the Twitter profile of the series was really valuable because of the data.

Transmedia communication, not only through the audiovisual development of the series, is a key element for Money Heist. In this sense, it is appropriate to make a series of recommendations that other shows can take into account when considering these transmedia strategies:

1. Transmedia strategies must be included in the initial marketing and communication strategy concept. They are, nowadays, a basic element.
2. These strategies must be based on different languages and different platforms. It is not only a question of transmitting the same message with the same language on different platforms
3. Even the strategy within the same social network, such as Twitter, should not be univocal. The different audiovisual languages must be understood in order to develop strategies in those languages that are considered relevant: video, audio, photos, gifs, or tweet threads.
4. In the specific case of Twitter, the dialogue can be improved. The creation of accounts related to characters that interact with each other and with the main user will make the transmedia strategy more realistic.

**Author Contributions:** Conceptualization: C.C.R. and P.M.N.; data curation: F.J.C.; formal analysis: F.J.C.; methodology: F.J.C.; project administration: C.C.R.; resources: F.J.C. and P.M.N.; software: F.J.C.; validation: C.C.R. writing—original draft, C.C.R., P.M.N. and F.J.C.; writing—review and editing, C.C.R. and F.J.C. All authors have read and agreed to the published version of the manuscript.

**Funding:** This research received no external funding.

**Conflicts of Interest:** The authors declare no conflict of interest

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
