# Peer review of "Release of the Fourth Season of Money Heist: Analysis of Its Social Audience on Twitter during Lockdown in Spain"

_information, doi:10.3390/info11120579_

Round 1

Reviewer 1 Report

There are some comments to be taken into consideration during the review process. Please see the file attached.

We also suggest tp add some limits of this article.

Author Response

Dear Reviewer,

We are really grateful for your comments. We have made the changes you requested in your thorough review. We believe that our work has improved with your input. Greetings

Reviewer 2 Report

The work is very interesting and current. The literature could be increased. The methodology could also include other aspects for further study; for example, the communication adopted on Facebook could also have been studied. In fact, on Facebook the length of the contents is greater and could have returned relevant information also with reference to the sentiment. Always for further developments, it would be interesting to know the user's perspective better. Interaction with content that also includes other media is always greater than with text-only content, regardless of the pandemic situation. The conclusions could also be deepened, especially as regards the managerial implications.

Author Response

(The authors gave the same response as above.)

Reviewer 3 Report

The article analyzes a topic of great interest for research on consumer habits for audiovisual products at the time of the epidemic. Observing, in particular, how such habits and consumption styles of leisure have changed, how the producers of platforms have been able to intercept the new consumer needs in a time of forced immobility at home and, finally, the increase (also quantitative, as well as qualitative) of the old and new media circuit. Starting from a specific case (Netflix and the successful Casa de Papel series), the author uses this case to argue the innovation that the Netflix platform is producing during pandemic times compared to the traditional audiovisual consumption of other televisual products. The author demonstrates a high knowledge of communication processes that expand from a television series to the communication chain created thanks to social media. The use of methodological tools is rigorous and the results produced by empirical research (combining analysis of discourse, analysis of "sentiments" and in-depth interviews) are of undoubted originality.

Author Response

(The authors gave the same response as above.)

Reviewer 4 Report

In this paper, authors did analysis about ‘Money Heist’ using its audience data on Twitter during lockdown in Spain. However, I have some suggestion for paper improvement as follows.

  1. The abstract is not properly written. Authors should make it more attractive.
  2. The introduction section is very lengthy and difficult to understand the novelty. I suggest to divide it in two section: Introduction and Related Work (discussed the existing work regarding analysis of Twitter data). Authors should also discuss existing research work about Twitter data mining (An intelligent healthcare monitoring framework using wearable sensors and social networking data, An Exploratory Data Analysis of the #Crowdfunding Network on Twitter).
  3. The authors have not included a single figure. It is better to represent the methodology in the form of figure.
  4. The methodology section must be extended by including more details.
  5. I am confused about results. Authors should include details about the results section (before starting section 4.1). Did the authors use some approaches (Machine learning or feature extraction….) to handle Twitter data?
  6. Where is conclusion? Authors should include conclusion.

Author Response

Dear Reviewer,

We are really grateful for your comments. We have made the changes you requested in your thorough review. We believe that our work has improved with your input.

- The wording of the Abstract has been modified.
- The introduction has been extended, we have improved in the elements you requested.
- We have included a flowchart that helps to understand the work done in terms of data analysis and processing.
- We have differentiated the discussion and conclusions, including some recommendations.

Round 2

Reviewer 4 Report

This paper can be accepted in its present form.